# Long non-coding RNAs as a source of new peptides

**Jorge Ruiz-Orera[1], Xavier Messeguer[2], Juan Antonio Subirana[1,3], M Mar Alba[1,4]***

[1]Evolutionary Genomics Group, Research Programme on Biomedical Informatics, Hospital del Mar Research Institute, Universitat Pompeu Fabra, Barcelona, Spain; [2]Llenguatges i Sistemes Informàtics, Universitat Politècnica de Catalunya, Barcelona, Spain; [3]Real Academia de Ciències i Arts de Barcelona, Barcelona, Spain; [4]Catalan Institution for Research and Advanced Studies, Barcelona, Spain

**Abstract** Deep transcriptome sequencing has revealed the existence of many transcripts that lack long or conserved open reading frames (ORFs) and which have been termed long non-coding RNAs (lncRNAs). The vast majority of lncRNAs are lineage-specific and do not yet have a known function. In this study, we test the hypothesis that they may act as a repository for the synthesis of new peptides. We find that a large fraction of the lncRNAs expressed in cells from six different species is associated with ribosomes. The patterns of ribosome protection are consistent with the translation of short peptides. lncRNAs show similar coding potential and sequence constraints than evolutionary young protein coding sequences, indicating that they play an important role in de novo protein evolution.

## Introduction

Studies performed over the past decade have unveiled a richer and more complex transcriptome than was previously appreciated (*Okazaki et al., 2002*; *Carninci et al., 2005*; *Kapranov et al., 2007*; *Ponjavic et al., 2007*). Thousands of long RNA molecules (>200 nucleotides) that do not display the typical properties of well-characterized protein-coding RNAs, and which have been named intergenic or long non-coding RNAs (lncRNAs), have been discovered in several eukaryotic genomes (*Okazaki et al., 2002*; *Ponting et al., 2009*; *Cabili et al., 2011*; *Liu et al., 2012*; *Pauli et al., 2012*; *Ulitsky and Bartel, 2013*). There are several lncRNAs that have regulatory functions (*Guttman and Rinn, 2012*; *Ulitsky and Bartel, 2013*). For example the X-inactive-specific transcript *Xist* regulates X chromosome inactivation in eutherian mammals (*Brockdorff et al., 1992*). However, the vast majority of lncRNAs do not have a known function.

Intriguingly, several recent studies have noted that a large fraction of lncRNAs associate with ribosomes (*Ingolia et al., 2011*; *Bazzini et al., 2014*; *Juntawong et al., 2014*; *van Heesch et al., 2014*). Deep sequencing of ribosome-protected fragments, or ribosome profiling, provides detailed information on the regions that are translated in a transcript (*Ingolia, 2014*). According to some studies, the patterns of ribosome protection indicate that lncRNAs are capable of translating short peptides (*Ingolia et al., 2011*; *Bazzini et al., 2014*; *Juntawong et al., 2014*) although others have reached different conclusions (*Guttman et al., 2013*). Many lncRNAs have the same structure as classical mRNAs: they are transcribed by polymerase II, capped and polyadenylated, and accumulate in the cytoplasm (*van Heesch et al., 2014*). However, in contrast to typical protein-coding genes, they tend to contain few introns, are expressed at low levels, exhibit weak sequence constraints, and show limited phylogenetic conservation (*Cabili et al., 2011*; *Derrien et al., 2012*; *Kutter et al., 2012*; *Necsulea et al., 2014*).

The association of lncRNAs with ribosomes, and the fact that many of them appear to have arisen relatively recently in evolution, indicate that they could be an important source of new peptides.

**\*For correspondence:** malba@ imim.es

**Competing interests:** The authors declare that no competing interests exist.

**Reviewing editor**: Diethard Tautz, Max Planck Institute for Evolutionary Biology, Germany

**eLife digest** Despite the terms being largely interchangeable in modern language, 'DNA' and 'gene' do not mean the same thing. A gene is made of DNA and contains the instructions to make a protein, and it is the protein that performs the function of the gene. However, cells in the body also contain DNA that does not form genes. Far from being 'junk' DNA with no biological purpose; this DNA has a variety of roles, including affecting how other genes are used.

To produce a protein, the DNA sequence of a gene is transcribed into an intermediate molecule called RNA, which is then translated to produce a protein. So-called long non-coding RNA (lncRNA) molecules are also transcribed from DNA, but whether these are translated to make proteins has been a subject of much debate. Indeed, the function of the vast majority of lncRNA molecules is unknown.

Ruiz-Orera et al. analyzed RNA sequences collected from earlier experiments on six different species—humans, mice, fish, flies, yeast, and a plant—and found nearly 2500 as yet unstudied lncRNAs in addition to those previously identified. Many of the lncRNAs that Ruiz-Orera et al. investigated could be found lodged inside the cellular machinery used to translate RNA into proteins. Furthermore, these lncRNA molecules are oriented in the machinery as if they are primed and ready for translation, suggesting that many lncRNAs do produce proteins. However, it is unclear how many of these proteins have a useful function.

Very few lncRNAs were found in more than one species, suggesting that they have evolved recently. The properties of lncRNA molecules also show many similarities with the properties of 'young'—recently evolved—genes that are known to produce proteins. The combined findings of Ruiz-Orera et al. therefore suggest that lncRNAs are important for developing new proteins. The emergence of proteins with new functions has been an important driving force in evolution, and this work provides important clues into the first steps of this process.

Levine et al., who described the first examples of de novo originated genes in *Drosophila melanogaster*, already noted that non-coding RNAs expressed at low levels could contribute to the birth of novel protein coding genes (*Levine et al., 2006*). Cai et al. found a new protein coding gene in *Saccharomyces cerevisiae* likely to have been formed from a previously transcribed non-coding sequence (*Cai et al., 2008*). Wilson and Masel observed that ribosome profiling reads from a yeast experiment often mapped to intergenic transcripts (*Wilson and Masel, 2011*), and they proposed that this could help provide the raw material for the birth of new protein-coding genes. Another study in yeast found evidence of translation of short species-specific ORFs located in non-genic regions (*Carvunis et al., 2012*). More generally, it is important to consider that de novo protein-coding gene evolution, which was once thought to be a very rare event, is now believed to be relatively common (*Khalturin et al., 2009*; *Toll-Riera et al., 2009*; *Tautz and Domazet-Lošo, 2011*; *Long et al., 2013*; *Reinhardt et al., 2013*). Recently emerged proteins tend to be very short and evolve under weak evolutionary constraints (*Albà and Castresana, 2005*; *Levine et al., 2006*; *Cai et al., 2009*; *Liu et al., 2010*; *Xie et al., 2012*; *Palmieri et al., 2014*), properties that we also expect to find in the putative ORFs of lncRNAs.

The idea that lncRNAs serve as a repository for the evolution of new peptides is appealing but the evidence is still fragmented. In this study, we have analyzed ribosome profiling experiments performed in six different species and measured the sequence coding potential and selective constraints of the putatively translated ORFs in lncRNAs and codRNAs. We have discovered that lncRNAs show very similar characteristics to evolutionary young protein coding genes (lineage-specific proteins). The results strongly support a role for lncRNAs in the production of new peptides.

## Results

### Characterization of coding and long non-coding transcripts

We obtained polyA+ RNA and ribosome profiling sequencing data from six different published experiments performed in diverse eukaryotic species, mouse (*Mus musculus*), human (*Homo sapiens*, HeLa cells), zebrafish (*Danio rerio*), fruit fly (*D. melanogaster*), *Arabidopsis* (*A. thaliana*), and yeast

**Table 1.** Data sets used in the study

| Species | | GEO Accession | Mapped reads (millions) | Max read length (bp) | Description | Reference |
|---|---|---|---|---|---|---|
| Mouse *M. musculus* | RNA-seq | GSE30839 | 226.0 | 43 | ES cells, E14 | *Ingolia et al., 2011* |
| | Ribosome profiling | GSE30839 | 39.2 | 47 | | |
| Human *H. sapiens* | RNA-seq | GSE22004 | 29.8 | 36 | HeLa cells | *Guo et al., 2010* |
| | Ribosome profiling | GSE22004 | 78.3 | 36 | | |
| Zebrafish *D. rerio* | RNA-seq | GSE32900 | 1382.2 | 2 × 75 | Series of developmental stages | *Chew et al., 2013* |
| | Ribosome profiling | GSE46512 | 1040.0 | 44 | | |
| Fruit fly *D. melanogaster* | RNA-seq | GSE49197 | 1317.9 | 50 | 0–2hr embryos, wild type | *Dunn et al., 2013* |
| | Ribosome profiling | GSE49197 | 105.7 | 50 | | |
| Arabidopsis *A. thaliana* | RNA-seq | GSE50597 | 79.8 | 51 | No stress conditions, TRAP purification | *Juntawong et al., 2014* |
| | Ribosome profiling | GSE50597 | 140.3 | 51 | | |
| Yeast *S. cerevisiae* | RNA-seq | GSE52119 | 20.54 | 50 | GSY83, diploid | *McManus et al., 2014* |
| | Ribosome profiling | GSE52119 | 6.83 | 50 | | |

(*S. cerevisiae*) (*Table 1*). After read mapping and transcript assembly, we classified the expressed transcripts longer than 200 nucleotides into coding and long non-coding classes (codRNAs and lncRNAs, respectively, *Table 2*).

We detected hundreds of annotated lncRNAs in the vertebrate species (mouse, human and zebrafish), the number being lower (<150) in the other species (fruit fly, *Arabidopsis* and yeast). In addition, we identified a large number of novel lncRNAs not annotated in the databases, 2488 taking all species together (*Supplementary file 1A*). The inclusion of such lncRNAs resulted in a sixfold increase in the number of lncRNAs amenable for study in zebrafish and a twofold increase in mouse. In yeast, we only found two annotated lncRNAs, but there were 19 novel ones. In the majority of the analyses, we merged the annotated and the novel lncRNAs.

As expected, lncRNAs tended to be much shorter than codRNAs in all the species studied (*Figure 1A*). We found that most lncRNAs contained at least one short ORF (≥24 amino acids) and often several ORFs. The average ORF size in lncRNAs was between 43 and 68 amino acids depending on the species (*Supplementary file 1B*). Consistent with previous studies, lncRNAs were expressed at significantly lower levels than codRNAs (*Figure 1B*, Wilcoxon test, $p < 10^{-5}$).

## Efficient detection of translation events by ribosome profiling

The analysis of ribosome profiling sequencing data showed that the percentage of expressed coding transcripts associated with ribosomes was >90% in all species, with the highest values (>99%) in mouse and fruit fly (*Table 2*). Pseudogenes had a lower rate of association with ribosomes than coding RNAs, but surprisingly, in species with many annotated pseudogenes, such as human, mouse, and *Arabidopsis*, the majority of them showed association with ribosomes (*Supplementary file 1A*). This appeared to be a true signal; while pseudogenes will typically show sequence similarity to other functional copies in the genome, we only considered uniquely mapped reads with no mismatches.

Ribosome profiling is based on deep sequencing, and thus provides an unmatched level of resolution of the translated peptides when compared with current proteomics techniques. This is especially important for short proteins, which are difficult to detect by standard mass spectrometry methods (*Slavoff et al., 2013*). We used the ribosome-associated protein-coding RNA data to investigate the relationship between peptide detection by proteomics and protein length. We found that human and mouse translated proteins between 24 and 80 amino acids long were more difficult to identify in proteomics databases than longer proteins (*Table 3*).

**Table 2.** Fraction of transcripts associated with ribosomes

| | codRNA | | | lncRNA | | |
| | Expressed | Associated with ribosomes (RP) | | Expressed | Associated with ribosomes (RP) | |
| | | Total | Stringent | | Total | Stringent |
|---|---|---|---|---|---|---|
| Mouse | 14,245 | 14,196 (99.7%) | 13,918 (97.7%) | 476 | 390 (81.9%) | 367 (77.1%) |
| Human | 17,011 | 16,630 (97.8%) | 16,617 (97.7%) | 934 | 403 (43.1%) | 343 (36.7%) |
| Zebrafish | 12,595 | 11,643 (92.4%) | 11,637 (92.4%) | 2392 | 726 (30.4%) | 684 (28.6%) |
| Fruit fly | 8041 | 8031 (99.9%) | 7623 (94.8%) | 28 | 22 (78.6%) | 10 (35.7%) |
| Arabidopsis | 19,162 | 18,879 (98.5%) | 10,329 (53.9%) | 139 | 93 (66.9%) | 68 (48.9%) |
| Yeast | 4740 | 4547 (95.9%) | 4335 (91.5%) | 21 | 6 (28.6%) | 6 (28.6%) |

Stringent: number of transcripts significant at $p < 0.05$ using 3′UTRs as a null model (see 'Materials and methods' for more details).

## Long non-coding RNA transcripts frequently associate with ribosomes

The percentage of lncRNAs scanned by ribosomes (lncRNA_ribo) was surprisingly high in all the species studied (*Table 2*). The values ranged from 28.6% in yeast to 81.9% in mouse. This affected the main lncRNA classes described in Ensembl v. 70, including long intervening non-coding RNAs (lincR-NAs) or antisense transcripts (*Supplementary file 1C*). Short transcript size may hinder ribosome association detection (*Aspden et al., 2014*). We also found that the ribosome profiling signal was more difficult to detect in poorly expressed transcripts than in highly expressed ones, both for lncR-NAs and codRNAs (*Figure 2*). As lncRNAs tend to be expressed at low levels and are short when compared to codRNAs (*Figure 1*), we might be underestimating their association with ribosomes.

In order to determine if the ribosome profiling signal in lncRNAs was different from noise, we compared ribosome density in the transcripts it to that in 3'untranslated regions (3'UTRs). More specifically, the null model consisted in a size-matched set of sequences containing randomly taken 3'UTR from annotated coding transcripts. Ribosome density was calculated as the number of ribosome profiling reads divided by RNA-seq reads, a ratio defined as translational efficiency (TE) (*Ingolia et al., 2011*). Both codRNAs and lncRNAS displayed much higher TE values than 3'UTRs in all species studied (Wilcoxon test $p < 10^{-5}$, *Figure 3*). We could reject the null model for 90.12% of the lncRNAs and 87.19% of the codRNAs associated with ribosomes ($p < 0.05$) (see details by species in *Table 2*, Stringent set). Therefore, we concluded that the density of ribosomes in lncRNAs is much higher than expected by spurious ribosome binding.

Next, we compared ribosome density in lncRNAs and codRNAs in each of the species focusing on regions covered by ribosome profiling reads to accommodate for any differences in the length of the putatively translated regions. In human, fruit fly, and yeast, TE was higher in codRNAs than in lncRNAs (Wilcoxon test, $p < 0.005$), but in mouse and zebrafish the opposite trend was observed (Wilcoxon test, $p < 0.05$) (*Figure 4*). Despite the differences between the species, which may be due to technical issues, it is clear that lncRNAs can show TE values that are similar or even higher than codRNAs. The results were similar when we restricted the analysis to genes encoding a single transcript to avoid any possible biases due to multiple read mapping or when we employed the maximum TE in 90 nucleotide windows (*Figure 4—figure supplement 1*).

For comparison, we collected a set of 29 human genes with non-coding functions described in several recent reviews (*Supplementary file 2A*; *Ponting et al., 2009*; *Ulitsky and Bartel, 2013*; *Fatica and Bozzoni, 2014*). Many of these genes play roles in the regulation of gene expression in the nucleus and are thus unlikely to be translated. We only detected expression for five of these genes: *Malat1*, *Pvt1*, *Neat1*, *Meg8*, and *Cyrano*. Transcripts encoded by the first three genes showed ribosome association. In the case of *Malat1*, this was also consistently observed in mouse and zebrafish (in the latter species *Malat1* was identified as a novel transcript) and in the case of *Pvt1* in mouse. Given the small number of expressed transcripts, we could not draw any general conclusions for this set.

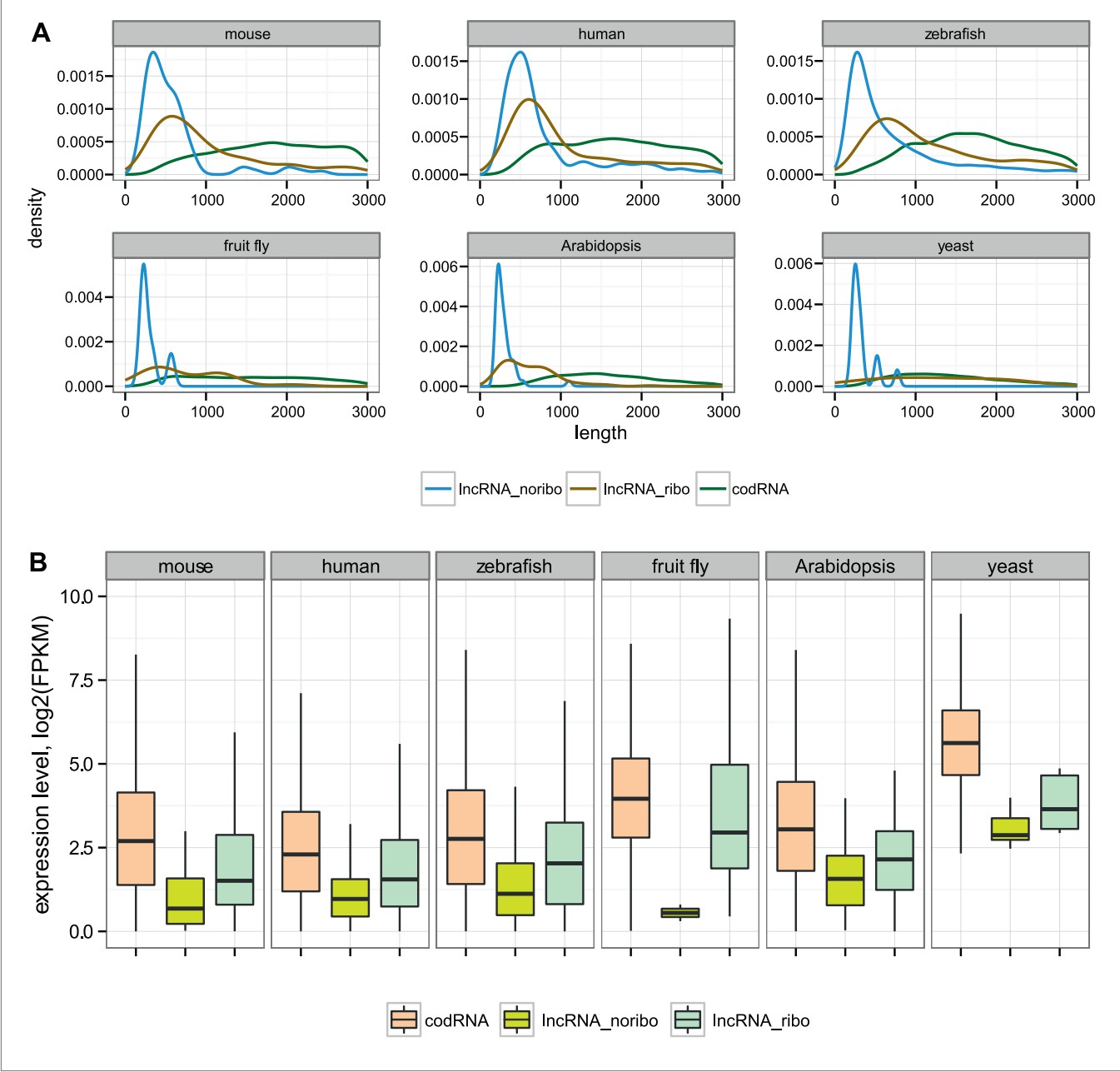

**Figure 1**. General characteristics of codRNA and lncRNA transcripts. (**A**) Density plots of transcript length. (**B**) Box-plots of transcript expression level in log2(FPKM) units. lncRNA_ribo: lncRNAs associated with ribosomes; lncRNA_noribo: lncRNAs for which association with ribosomes was not detected. codRNA: coding transcripts encoding experimentally validated proteins except for zebrafish in which all transcripts annotated as coding were considered. The area within the box-plot comprises 50% of the data and the line represents the median value. In all studied species, codRNAs were expressed at higher levels than lncRNAs (Wilcoxon test, $p < 10^{-5}$), and lncRNA_ribo at higher levels than lncRNA_noribo (Wilcoxon test, $p < 0.005$).

## lncRNAs show similar ribosome protection profiles to codRNAs

The exact positions of ribosome profiling reads on the RNA can be used to delineate the regions that are being actively translated or to discover new functional ORFs (*Chew et al., 2013*; *Guttman et al., 2013*; *Ingolia, 2014*). Because the ribosome is released after encountering a stop codon, this technique can also be employed to identify novel C-terminal protein extensions (*Dunn et al., 2013*) or to evaluate if a predicted ORF is likely to correspond to a translated peptide (*Guttman et al., 2013*). We next aimed at comparing the TE values in different transcript regions, including open reading frames

**Table 3.** Fraction of translated proteins of different size detected in proteomics databases

| Species | Protein size (amino acids) | | | |
| --- | --- | --- | --- | --- |
| | 24–80 | 81–130 | 131–180 | >180 |
| Mouse | 27/58 (46.6%) | 222/286 (77.6%) | 256/330 (77.6%) | 3716/4786 (77.7%) |
| Human | 116/272 (42.6%) | 536/748 (71.7%) | 669/875 (76.5%) | 6757/8964 (75.4%) |
| Yeast | 27/30 (90.0%) | 168/207 (81.1%) | 234/265 (88.3%) | 2934/3224 (91.0%) |

Only transcripts encoding experimentally validated proteins (codRNAe) were considered.

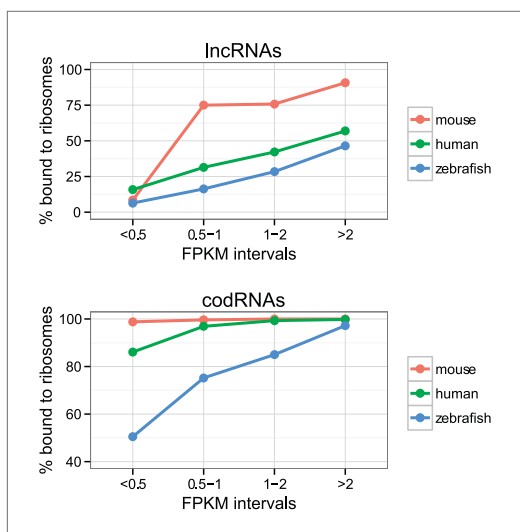

**Figure 2**. Effect of transcript expression level on the detection of ribosome association. The percentage of transcripts associated with ribosomes is shown for several transcript expression intervals. codRNA: annotated coding transcripts encoding experimentally verified proteins (except in zebrafish for which all coding transcripts were considered). lncRNA: annotated and novel long non-coding RNAs. Only species with at least 20 transcripts in each expression bin were plotted. In the rest of species, the data were consistent with the trends shown.

(ORFs), putative 5′ and 3′ untranslated regions (UTRs), and the regions between ORFs.

In order to obtain an unbiased picture, it was important to define the different regions in the same way in lncRNAs and codRNAs. In typical codRNAs there is a main translated ORF that covers a large fraction of the transcript, sometimes accompanied by short upstream ORFs in the 5′UTR (*Chew et al., 2013*). However, lncRNAs may potentially encode several short peptides (*Ingolia et al., 2011*). The minimum size of ORFs was set at 24 amino acids (75 nucleotides counting the STOP codon), as peptides of this size have been identified in genetic screen studies in humans (*Hashimoto et al., 2001*). To simplify the comparisons, we employed the same ORF size cut-off in all species. We also considered both a primary ORF, defined as the ORF with the largest number of ribosome profiling reads, as well as any additional non-overlapping ORFs that mapped to ribosome profiling reads (rest of ORFs).

In codRNAs, the primary ORF showed a nearly perfect degree of agreement with the annotated protein, indicating that it was an appropriate metric for the main translated product. Primary ORFs in lncRNAs typically occupied a shorter fraction of the transcript than in codRNAs (*Figure 5A*). The relative length of the ORF with respect to transcript length did not seem to be a strong predictor of ribosome association, as it did not help distinguish lncRNAs associated with ribosomes (lncRNA_ribo) to those not associated with ribosomes (lncRNA_noribo). In lncRNAs, most of the primary ORFs corresponded to proteins less than 100 amino acids long (*Figure 5—figure supplement 1*).

Next, we focused our attention on the differences between the primary ORF and the 5′UTR and 3′UTR regions in codRNAs and lncRNAs. We defined the 3′ untranslated region (3′UTR) as the sequence located immediately after the STOP codon of the primary ORF or the most downstream ORF associated with ribosomes. We used the same criteria to define the 5′UTR upstream from the initiation codon. In this analysis, we included all transcripts containing at least one ORF associated with ribosomes (the primary ORF) and sufficiently long UTR regions as to detect ribosome profiling reads (>30 nucleotides); insufficient data for fruit fly and yeast precluded the analysis for these species. In both codRNAs and lncRNAs, the 5′UTR showed a ribosome density (translational efficiency, TE) comparable to that of the primary ORF (*Figure 5B*). In contrast, the 3′UTR showed very little ribosome association and often we could not find a single read mapping to this region (31–91% of cases in codRNAs and 46–68% in lncRNAs). Using genes with a single isoform or considering only annotated

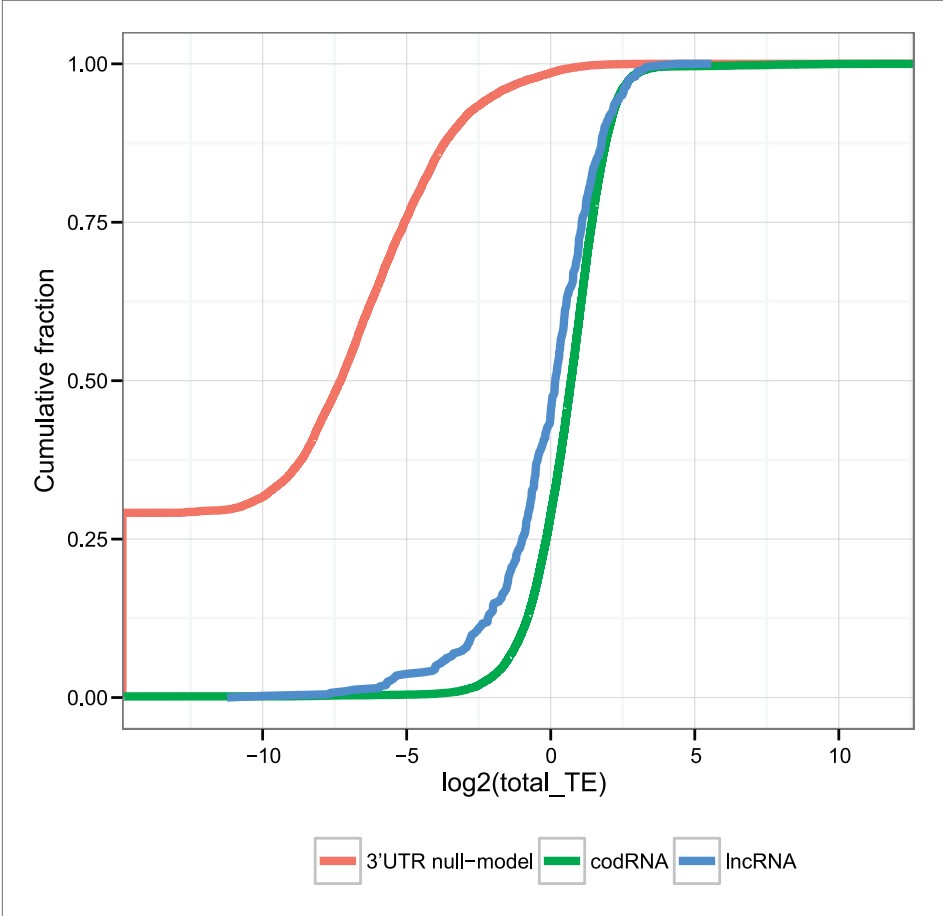

**Figure 3**. TE distribution in human transcripts and 3'UTRs (null-model). Cumulative distribution of TE values in human codRNAs, lncRNAs, and 3'UTR sequences. We randomly selected 3'UTRs with a minimum length of 30 nucleotides to build a set of 3'UTR sequences with the same size distribution as the complete transcripts.

transcripts produced similar results (*Figure 5—figure supplements 2 and 3*). We also controlled for expression level by dividing the data set in transcripts with low (0.5–2 FPKM) and high expression (>2 FPKM), and by sampling the codRNAs in such a way as to have a similar expression distribution as lncRNAs. The results were very similar to those obtained with the complete data set (*Figure 5—figure supplement 4*), indicating that the analysis is robust to transcript expression differences.

As transcripts may contain several ORFs, we performed a separate analysis in which we compared the translational efficiency of the primary ORF, any additional ORFs with mapped ribosome profiling reads, and the regions between ribosome-protected ORFs (interORF) (*Figure 5C*). InterORF regions showed little signal when compared to the primary ORF, both in codRNAs and lncRNAs (Wilcoxon test, $p < 10^{-9}$ in human, mouse, and zebrafish, $p < 0.05$ in *Arabidopsis*, insufficient data for fruit fly and yeast precluded the analysis for these species). The data also indicated that ribosome binding is not always restricted to the primary ORF, especially in lncRNAs, as ribosome protection could sometimes be observed for additional ORFs.

Taken together, these results indicate that lncRNAs have ribosome profiling signatures consistent with translation, with a strong decrease of ribosome density in the 3'UTR but not the 5'UTR region, and preferential binding of ribosomes to the primary ORF. There exists the possibility that the translated peptides are degraded soon after being produced. However, we estimate that the percentage of cases that may undergo nonsense-mediated decay (NMD, see 'Materials and methods' for more details) is low, between 4.47 and 14.11% depending on the species. For comparison, the percentage for protein-coding transcripts showing the same patterns (including transcripts annotated as NMD in Ensembl) is between 0.34 and 13.33%.

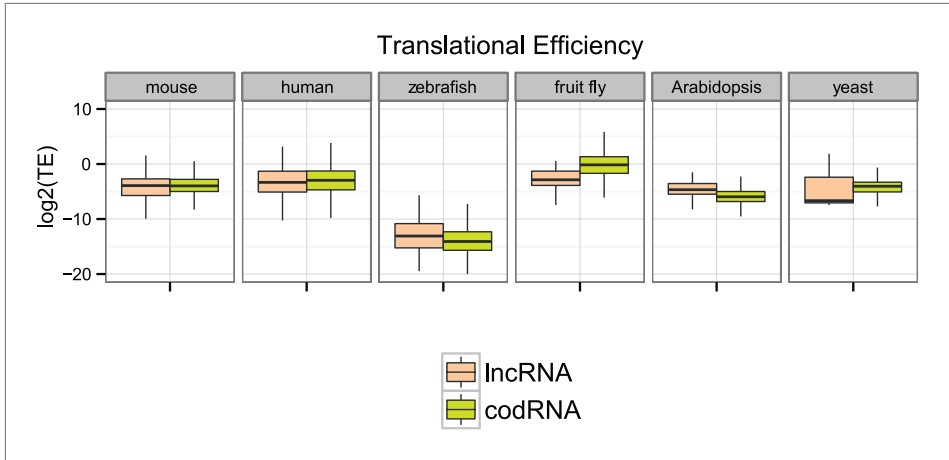

**Figure 4**. Ribosome association profiles for codRNAs and lncRNAs. Box-plots of transcript translational efficiency (TE) in log2(TE) units. The area within the box-plot comprises 50% of the data, and the line represents the median value. lncRNA: lncRNAs for which association with ribosomes was detected. codRNA: coding RNAs transcripts encoding experimentally validated proteins except for zebrafish in which all transcripts annotated as coding were considered.

The following figure supplement is available for figure 4:

**Figure supplement 1**. Additional translational efficiency (TE) measures.

## lncRNAs are less conserved than codRNAs

Are the putatively translated ORF in lncRNAs conserved? We performed sequence similarity searches using BLASTP (E-value < $10^{-4}$) against all annotated coding transcripts in Ensembl, as well as against the primary ORFs in lncRNAs, for the six species studied here (*Supplementary files 1D and 2B*). The number of lncRNA_ribo with homologues in other species was remarkably low (0–15.6%) except for zebrafish (49.4%). In contrast, the majority of codRNAs had homologues in other species (>95% for vertebrates and fruit fly and 70–73% for *Arabidopsis* and yeast). After we discarded lncRNAs that showed cross-species conservation, association with ribosomes was still very prevalent (80.4% of mouse, 40.3% of human, and 22.1% of zebrafish lncRNAs were associated with ribosomes).

We also investigated whether the ribosome-associated ORFs in lncRNAs showed homology to annotated proteins in the same species. The values were very low for all the species (0–12.4%) except for zebrafish (47.5%). Therefore, in general lncRNAs are not truncated duplicated copies (pseudogenes). The case of zebrafish is an exception probably because of missing protein-coding annotations in this species.

## Coding properties of ribosome-protected ORFs in lncRNAs

Subsequently, we compared the sequence coding properties of the primary ORF in lncRNAs with those in *bona fide* coding and non-coding sequences using a hexamer-based coding score (see 'Materials and methods'). In all species the coding scores of the primary ORF in lncRNAs, while lower than that of codRNAs, were significantly higher than the coding score of ORFs in introns (*Figure 6*, Wilcoxon test lncRNA_ribo vs intron, human, mouse, zebrafish, and *Arabidopsis* p < $10^{-16}$; fruit fly and yeast p < $10^{-5}$). This clearly shows that ORFs in lncRNAs are more coding-like than random ORFs. We repeated the same comparison using 100 different randomly sampled intronic sequence sets, and in >95% of the cases, we obtained the same result. lncRNAs associated with ribosomes (lncRNA_ribo) showed higher coding scores than those not associated with ribosomes (lncRNA_noribo), even when we did not use the ribosome profiling information and compared the longest ORF in both types of transcripts (*Figure 6—figure supplement 1*). We reached similar conclusions when we restricted the analysis to annotated lncRNA transcripts (*Figure 6—figure supplement 2*), when we used ORFs from gene deserts as an alternative non-coding sequence set (differences with lncRNAs significant by Wilcoxon test, p < $10^{-16}$, see 'Materials and methods' for more details), and when we restricted the

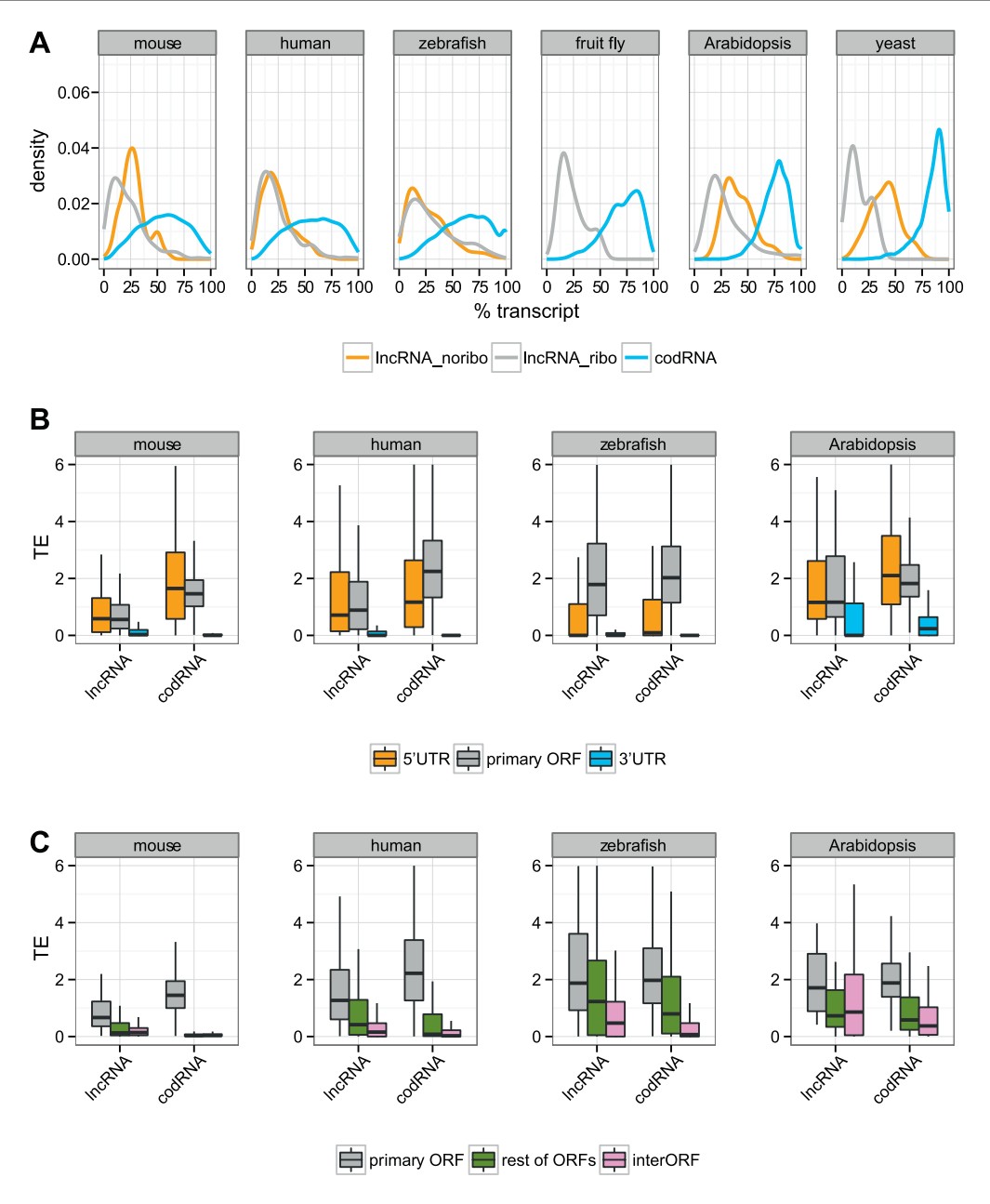

**Figure 5**. Ribosome association in different transcript regions. (**A**) Density plot of the relative length of the primary ORF in lncRNA_ribo and codRNA with respect to transcript length. For comparison data for the longest ORF in lncRNA_noribo is also shown (except for fruit fly due to insufficient data). (**B**) Box-plots of TE distribution in primary ORF, 5'UTR, and 3'UTR regions. The area within the box-plot comprises 50% of the data, and the line represents the median value. The analysis considered all transcripts with 5'UTR and 3'UTR longer than 30 nucleotides and >0.2 FPKM in all three regions. The number of transcripts was 1956 codRNA and 159 lncRNA_ribo in mouse, 3558 codRNA and 139 lncRNA_ribo in human, 5216 codRNA and 252 lncRNA_ribo in zebrafish, and 2019 codRNA and 33 lncRNA_ribo in Arabidopsis. (**C**) Box-plots of TE distribution in primary ORFs, rest of ORFs with ribosome profiling reads and non-ORF regions (interORF). The analysis considered all transcripts with at least two ORFs and more than 30 nucleotides interORF. The number of transcripts was 3264 codRNA and 204 lncRNA_ribo in mouse, 3104 codRNA and 168 lncRNA_ribo in human, 1646 codRNA and 212 lncRNA_ribo in zebrafish, and 1098 codRNA and 25 lncRNA_ribo in Arabidopsis. Fruit fly and yeast were not included in the last two analyses due to insufficient data (<8 lncRNA_ribo meeting the conditions).

*Figure 5. Continued on next page*

*Figure 5. Continued*

The following figure supplements are available for figure 5:

**Figure supplement 1**. Absolute nucleotide length of ORFs in different kinds of transcripts.

**Figure supplement 2**. Translational efficiency in single-isoform genes.

**Figure supplement 3**. Translational efficiency in annotated transcripts.

**Figure supplement 4**. Translational efficiency in transcripts expressed at different levels.

analysis to lncRNAs for which we did not find protein coding homologues in the other species studied (*Figure 6—figure supplement 3*). Because a high proportion of lncRNAs contained small ORFs, we repeated the comparison only considering transcripts with ORFs shorter than 100 amino acids to avoid any length biases, again obtaining similar results (*Figure 6—figure supplement 4*). The use of other coding scores, for example based on codon frequencies instead of hexamer frequencies or related metrics such as GC content produced consistent results (*Figure 6—figure supplement 5*; *Supplementary file 1E*).

At the individual transcript level, a sizeable fraction of lncRNAs associated with ribosomes displayed significantly higher coding scores than expected for non-coding sequences ($p < 0.05$ in all 100 intronic random sets; data in *Supplementary file 2C*; examples in *Figure 6—figure supplement 6*). These transcripts are comprised of 143 human lncRNAs (35.5% of the lncRNAs, score > 0.0189), 137 mouse lncRNAs (35.1%, score > 0.0377), 379 zebrafish lncRNAs (52.1% score > 0.0095), 7 fruit fly lncRNAs (31.8%, score > −0.0483), 43 *Arabidopsis* lncRNAs (46.2%, score > −0.0202), and 5 yeast lncRNAs (83.3%, score > 0.03387). Annotated and novel lncRNAs were present in similar proportions in these sets, supporting the validity of our strategy of merging the two types of transcripts from the beginning. We also noted that the fraction of lncRNAs with coding homologues in other species increased in these sets. For example, whereas the proportion of total human lncRNA_ribo with homologues in other species was 15.6%, in the set with significant coding scores it was 29.3%. This number increased to 57.3% when we performed searches against the NCBI non-redundant peptide database 'nr', as some of the ORFs in lncRNAs are annotated as predicted peptides in this database.

If ORFs in lncRNAs are being translated this is likely to be a relatively recent evolutionary event, as many lncRNAs are lineage-specific (*Pauli et al., 2012*; *Necsulea et al., 2014*; our data). It is well established that proteins of different evolutionary age display distinct sequence properties, including different codon usage (*Toll-Riera et al., 2009*; *Carvunis et al., 2012*; *Palmieri et al., 2014*). We retrieved sets of annotated protein-coding transcripts of different evolutionary age from human, mouse, zebrafish, *Arabidopsis*, and yeast available from various studies (*Ekman and Elofsson, 2010*; *Donoghue et al., 2011*; *Neme and Tautz, 2013*) and expressed in the systems studied here. We found that the coding score was always lower in the youngest group than in older groups (*Figure 6*, Wilcoxon test, $p < 0.05$). Remarkably, the youngest codRNAs showed a very similar coding score distribution to lncRNAs (*Figure 6*). We obtained similar results when we discarded lncRNAs that had homologues in any of the other species (*Figure 6—figure supplement 3*).

We also collected information from young protein coding genes encoding experimentally verified proteins according to Swiss-Prot (*Supplementary file 2D*). We observed that these proteins were short and the ORF occupied a relatively small fraction of the transcript, features typically observed in lncRNAs. For example, the average size of proteins encoded by primate-specific transcripts was 148 amino acids and the average transcript coverage 47%. The coding score was remarkably low and again similar to that of lncRNAs (median 0.008 for primate-specific human transcripts, 0.046 for rodent-specific mouse transcripts, and 0.089 for yeast-specific coding transcripts).

## Selection pressure signatures in ORFs associated with ribosomes

An important measure of the strength of purifying selection acting on a coding sequence is the ratio between the number of non-synonymous and synonymous single nucleotide polymorphisms (PN/PS). Given the nature of the genetic code, there are more possible non-synonymous mutations than

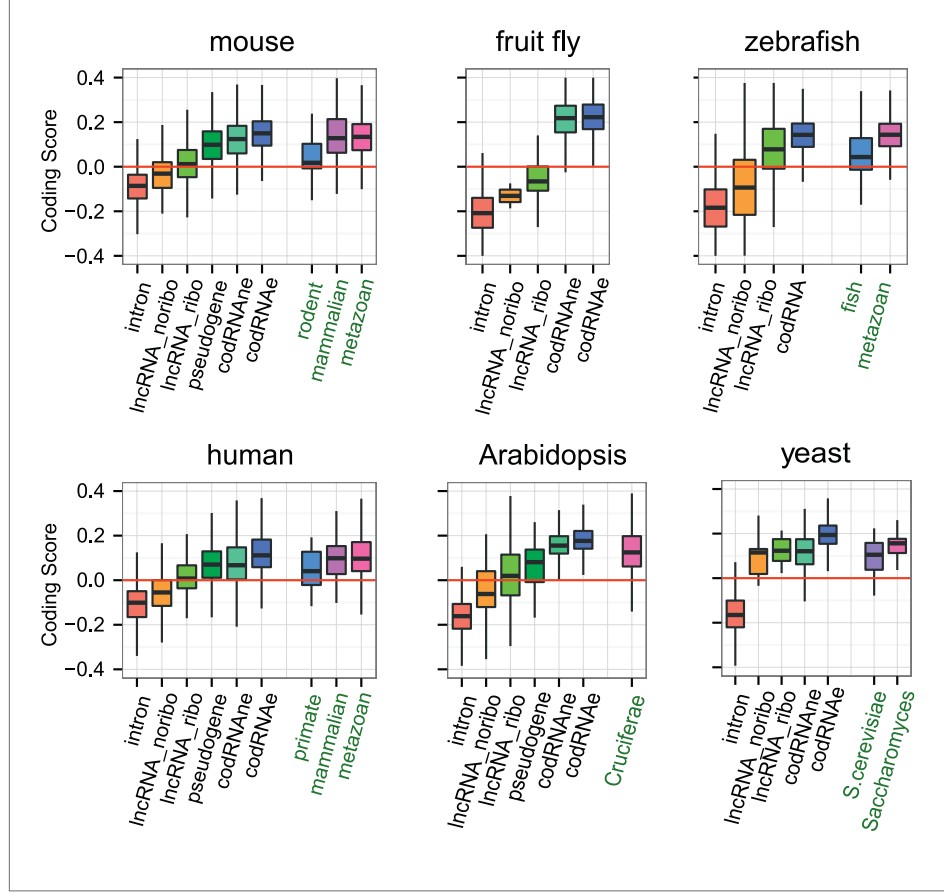

**Figure 6**. Coding scores in ORFs from different types of transcripts. Intron: randomly selected intronic regions; lncRNA_noribo: lncRNAs not associated with ribosomes; lncRNA_ribo: lncRNAs associated with ribosomes; pseudogene: pseudogenes associated with ribosomes; codRNAne: coding transcripts encoding non-validated proteins associated with ribosomes; codRNAe: coding transcripts encoding experimentally validated proteins. The coding score was calculated as the log ratio of hexamer frequencies in coding vs intronic sequences. In lncRNA_noribo and introns, we considered the longest ORF and in the rest of transcripts the primary ORF. The Class 'pseudogene' was only included in species with more than 20 expressed pseudogenes with mapped ribosome profiling reads. The coding score of the primary ORF in lncRNAs (lncRNA_ribo) was significantly higher than the coding score in ORFs defined in introns (Wilcoxon test, human, mouse, zebrafish, and Arabidopsis $p < 10^{-16}$; fruit fly and yeast $p < 10^{-4}$, Wilcoxon test) and in lncRNA_ribo it was significantly higher than in lncRNA_noribo in four species (Wilcoxon test, human, mouse and zebrafish $p < 10^{-5}$, and Arabidopsis $p < 0.05$). Transcripts from genes of different evolutionary age were taken from the literature (see manuscript text). The number of transcripts was 68 for rodent, 127/123 for mammalian (mouse/human as reference species), 11,203/13,423/9812 for metazoan (mouse/human/zebrafish), 162 for fish, 208 for Crucifera, 28 for *S. cerevisiae* and 84 for Saccharomyces. The youngest class of codRNAs displayed similar scores than lncRNA_ribo in mouse, zebrafish, and yeast (classes rodent, fish and *S. cerevisiae*, respectively), being only significantly higher in human and Arabidopsis (Wilcoxon test, $p < 0.005$; classes primate and Cruciferae). We did not analyze young genes in fruit fly due to lack of a suitable young set of codRNAs in this species.

The following figure supplements are available for figure 6:

**Figure supplement 1**. Coding scores for the longest ORF.

**Figure supplement 2**. Coding scores in different classes of annotated sequences.

**Figure supplement 3**. Coding scores in lncRNAs without homologues in other species.

*Figure 6. Continued on next page*

*Figure 6. Continued*

**Figure supplement 4**. Coding scores in small ORFs from different types of transcripts.

**Figure supplement 5**. Use of different coding statistics in human transcripts.

**Figure supplement 6**. Ribosome protection patterns in transcripts containing short ORFs.

synonymous mutations. Under neutrality (no purifying selection), the PN/PS ratio is expected to be approximately 2.89 (*Nei and Gojobori, 1986*).

Here, we applied the large amount of available polymorphism data for human, mouse, and zebrafish to compare the level of purifying selection in primary ORFs from codRNAs and lncRNAs (*Figure 7*; *Supplementary file 1F*). In general, human sequences showed higher PN/PS ratios than sequences from the other analyzed species, probably due to the presence of many slightly deleterious mutations segregating in the population (*Eyre-Walker, 2002*). However, despite the intrinsic differences between organisms, we observed the same general trends. First, the PN/PS was significantly lower in codRNAs than in lncRNAs (proportion test, $p < 10^{-5}$), denoting stronger purifying selection in the former. Second, there was a very clear inverse relationship between the strength of purifying selection and the age of the gene ($p < 10^{-15}$ between the youngest and rest of codRNAs in mouse and zebrafish), in agreement with previous studies (*Liu et al., 2008*; *Cai et al., 2009*). High PN/PS values were also observed in the subset of young genes encoding experimentally validated proteins in human (primate-specific transcripts median PN/PS of 3.10) and mouse (rodent-specific transcripts median PN/PS 1.42), confirming this tendency. Third, the distribution of PN/PS values in lncRNAs was very similar to that of young protein-coding genes. In human and mouse, there were no significant differences, and in the case of zebrafish the lncRNAs had even slightly lower PN/PS values than the fish-specific protein coding genes ($p < 0.01$).

## Discussion

Here, we analyzed the patterns of ribosome protection in polyA+ transcripts from cells belonging to six different eukaryotic species. Among the expressed transcripts, we identified many lncRNAs in the

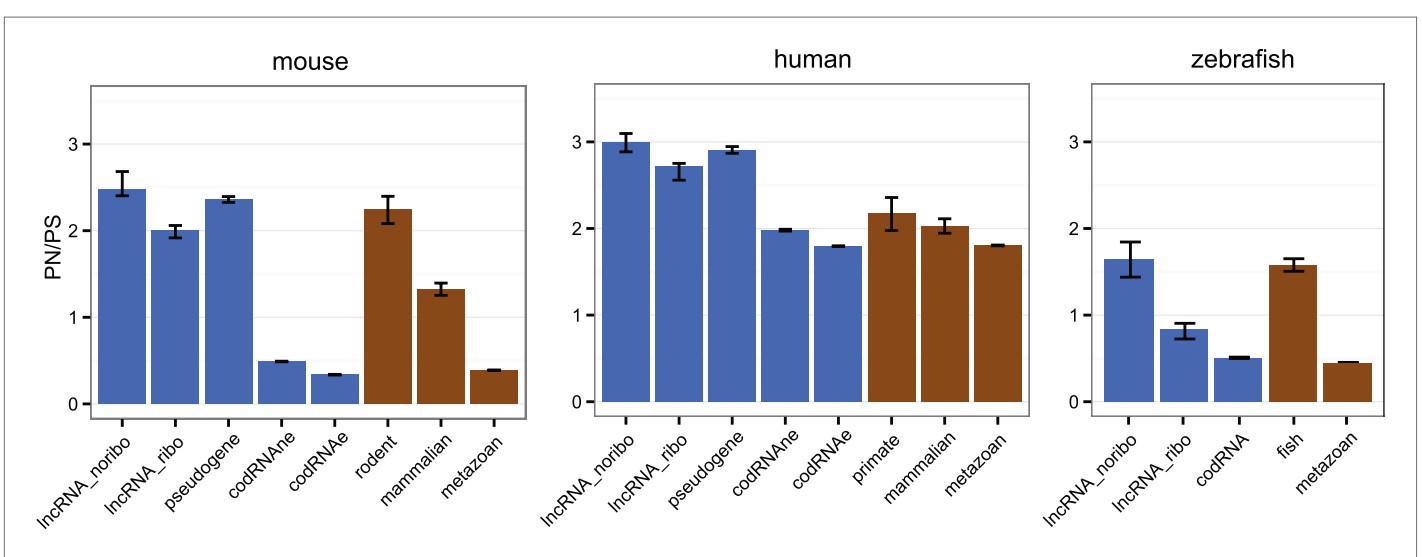

**Figure 7**. Selective pressure in ORFs from different types of transcripts. PN/PS: ratio between the number of non-synonymous and synonymous single nucleotide polymorphisms (SNPs) in the complete set of primary ORFs for a given class of transcripts (in lncRNA_noribo the longest ORF was considered). In blue, data for different coding and non-coding transcript classes. In brown, data for different age codRNA classes. The bars represent the 95% confidence interval for the PN/PS value. For the species not shown there was not sufficient data to perform this analysis.

different species. The vast majority of transcripts annotated as coding showed association with ribosomes (>92% in all species). Remarkably, a very large number of transcripts annotated as long non-coding RNA (lncRNAs) also showed such association (30–82% depending on the data set). Considering that lncRNAs are typically much shorter and expressed at lower levels than codRNAs, which may hinder the identification of ribosome association, this is a very significant fraction. In addition, the patterns of ribosome protection along the transcript are similar to those of protein-coding genes. Therefore, many lncRNAs appear to be scanned by ribosomes and are likely to translate short peptides.

Long non-coding RNAs are classified as such in databases because, according to a number of criteria, they are unlikely to encode functional proteins. These criteria include the lack of a long ORF, the absence of amino acid sequence conservation, and the lack of known protein domains (*Harrow et al., 2012*). Moreover, we expect lncRNAs not to have matches to proteomics databases, as this should classify them as coding. Annotated lncRNAs are typically longer than 200 nucleotides because this is the cutoff size normally implemented to differentiate them from other RNA classes such as microRNAs and small nuclear RNAs. In practice, it is difficult to classify a transcript as coding or non-coding on the basis of the ORF size (*Dinger et al., 2008*). Some true coding sequences may be quite small, and by chance alone non-coding transcripts may have relatively long ORFs. The majority of lncRNAs contain ORFs longer than 24 amino acids, which can potentially correspond to real proteins. Short proteins are more difficult to detect than longer ones and consequently they are probably underestimated in databases. In recent years, the use of comparative genomics (*Frith et al., 2006*; *Ladoukakis et al., 2011*; *Hanada et al., 2013*), proteomics (*Slavoff et al., 2013*; *Vanderperre et al., 2013*; *Ma et al., 2014*), and a combination of evolutionary conservation and ribosome profiling data (*Crappé et al., 2013*; *Bazzini et al., 2014*) have shown that the number of short proteins is probably much higher than previously suspected (*Andrews and Rothnagel, 2014*). In yeast, gene deletion experiments have provided evidence of functionality for short open reading frames (sORFs < 100 amino acids) (*Kastenmayer et al., 2006*); in zebrafish, several newly discovered sORFs appear to be involved in embryonic development (*Pauli et al., 2014*) and other examples exist in insects (*Magny et al., 2013*) and humans (*Lee et al., 2013*; *Slavoff et al., 2014*). In many cases, the transcripts containing sORFs will be classified as non-coding, especially if the ORF is not well conserved across different species.

One approach to identify potential coding transcripts is ribosome profiling (*Ingolia et al., 2009*), which has been used to study translation of proteins in a wide range of organisms (*Guo et al., 2010*; *Ingolia et al., 2011*; *Brar et al., 2012*; *Michel et al., 2012*; *Chew et al., 2013*; *Dunn et al., 2013*; *Huang et al., 2013*; *Artieri and Fraser, 2014*; *Bazzini et al., 2014*; *Juntawong et al., 2014*; *McManus et al., 2014*; *Vasquez et al., 2014*). In several of these studies it has been noted that lncRNAs can be protected by ribosomes (*Ingolia et al., 2011*; *Chew et al., 2013*; *Bazzini et al., 2014*; *Juntawong et al., 2014*). However, there is no consensus on whether the observed patterns are consistent with translation. For example in the original analysis of mouse stem cells, which we reanalyzed here, it was reported that many lncRNAs were polycistronic transcripts encoding short proteins (*Ingolia et al., 2011*), but in another paper where the same data were processed in a different way, they concluded that lncRNAs were unlikely to be protein-coding (*Guttman et al., 2013*). A zebrafish ribosome profiling study reported resemblance between lncRNAs and 5′leaders of coding RNAs; the authors suggested that translation may play a role in lncRNA regulation (*Chew et al., 2013*). Nevertheless, in the same study dozens of lncRNAs were proposed to be *bona fide* protein-coding transcripts. In *Arabidopsis*, the translational efficiency values of highly expressed lncRNAs (>5 FPKM) were similar to those of coding RNAs and some lncRNAs had profiles consistent with initiation and termination of translation (*Juntawong et al., 2014*). Finally, using yeast data, *Wilson and Masel. (2011)* found many cases of non-coding transcripts bound to ribosomes and suggested that this facilitates the evolution of novel protein-coding genes from non-coding sequences.

The disparity of results obtained in different systems motivated us to retrieve the original data and perform exactly the same analyses for six different species. As lncRNA catalogues are still very incomplete for most species, we also defined sets of novel lncRNAs using the RNA-seq sequencing reads for de novo transcript assembly. We discovered many novel, non-annotated, lncRNAs, especially in zebrafish, mouse, and fruit fly (*Table 2*). After the analysis of the ribosome profiling data, the same general picture emerged for the different biological systems, indicating that we are detecting very fundamental properties. In transcripts classified as lncRNAs, the ribosome profiling reads tend to cover a smaller fraction of the transcript than in typical codRNAs, in agreement with a shorter relative size of the ORF accumulating the largest number of ribosome profiling reads (primary ORF). We also find that

the translational efficiency of regions corresponding to the primary ORF is much higher than that of 3′UTRs, both in codRNAs and lncRNAs, consistent with translation of the transcripts. Furthermore, the primary ORF of lncRNAs showed significantly higher coding score than the longest ORF extracted from randomly selected non-coding regions.

lncRNAs often contain several potentially translated ORFs (*Ingolia et al., 2011*). Transcripts encoding multiple short proteins have been reported in insects (*Savard et al., 2006*) and could be common in other species as well (*Tautz, 2009*). One such candidate is AT1G34418.1 in *Arabidopsis*, an annotated lncRNA which contains a primary ORF followed by two instances of a 12 amino acid ORF also covered by ribosome profiling reads (*Figure 6—figure supplement 6*). This case is reminiscent of the gene *pri* in fruit fly, which regulates tarsal development (*Galindo et al., 2007*) and translates several small redundant ORFs (*Kondo et al., 2007*).

lncRNAs are poorly conserved across species and so, if translated, they will produce species- or lineage-specific proteins. Recently evolved proteins are markedly different from widely distributed ancient proteins; they are shorter, subject to weaker selective constraints and expressed at lower levels (*Albà and Castresana, 2005*; *Cai et al., 2009*; *Liu et al., 2010*; *Donoghue et al., 2011*; *Carvunis et al., 2012*; *Xie et al., 2012*; *Wissler et al., 2013*; *Neme and Tautz, 2014*). Here for the first time, we have compared the properties of the ORFs in lncRNAs associated with ribosomes with the properties of annotated, and in some cases experimentally validated, young protein-coding genes. lncRNAs and young protein-coding transcripts are virtually indistinguishable regarding their coding score and ORF selective constraints (*Figures 6 and 7*), which is consistent with the idea that many lncRNAs encode new peptides.

Although it is unclear how many of these peptides are functional, the data indicate that at least a fraction of them may be functional. Sequences that translate functional proteins are expected to display signs of selection related to preferential usage of certain amino acids and codons. This can be used to differentiate between coding and non-coding entities, especially in the absence of cross-species conservation, as is the case of many lncRNAs. About 35–40% of primary ORFs in human and mouse lncRNAs displayed coding scores that were significantly higher than those expected for non-coding sequences, making them excellent candidates for translating functional proteins. In fact, five human lncRNAs associated with ribosomes that exhibited high coding scores in our study were re-annotated as protein-coding transcripts in a subsequent Ensembl gene annotation release (version 75, *Supplementary file 2C*). Gene knock-out experiments in fly have discovered that young proteins, even if rapidly evolving, are often essential for the organism and can cause important defects when deleted (*Chen et al., 2010*; *Reinhardt et al., 2013*). Similarly, some peptides translated from lncRNAs may have important cellular functions yet to be discovered.

lncRNAs tend to be expressed at much lower levels than typical codRNAs, so, everything else being equal, the amount of translated peptide is also expected to be smaller. It may be that some of these peptides are not functional, but their translation does not produce a large enough deleterious effect for them to be eliminated via selection. Pseudogenes also showed extensive association with ribosomes in our study, indicating that the translation machinery is probably not very selective or that some pseudogenes produce functional proteins. This question may be worth revisiting, as a recent proteomics study has also found that dozens of human pseudogenes produce peptides (*Kim et al., 2014*).

The data also indicate that a fraction of lncRNAs have not acquired the capacity to be translated. Depending on the experiment analyzed, a number of lncRNAs did not show any significant association with ribosomes. As previously discussed, this is probably affected by a lack of sensitivity; it is also true that the lncRNAs not associated with ribosomes tended to show lower coding scores than lncRNAs associated with ribosomes, even when we did not use the ribosome profiling data and simply compared the longest ORF in both kinds of transcripts.

Recently, it has been reported that human-specific protein-coding genes are often related to non-coding transcripts in macaque, pointing to a non-coding origin for many newly evolved proteins (*Xie et al., 2012*). More generally, one may view de novo protein-coding gene evolution as a continuum from non-functional genomic sequences to fully-fledged protein-coding genes (*Albà and Castresana, 2005*; *Toll-Riera et al., 2009*; *Carvunis et al., 2012*). Therefore, many lncRNAs could be in intermediate states in this process, their pervasive translation serving as the building material for the evolution of new proteins. It may be difficult to obtain functional proteins from completely random ORFs (*Jacob, 1977*), but the effect of natural selection preventing the production of toxic

peptides (*Wilson and Masel, 2011*), and the high number of transcripts expressed in the genome, may facilitate this process.

## Materials and methods

### Sequencing and mapping of reads

We downloaded the original data from Gene Expression Omnibus (GEO) for six different ribosome profiling experiments that had both ribosome footprinting and polyA+ RNA-seq sequencing reads: mouse (*M. musculus*) (*Ingolia et al., 2011*), human (*H. sapiens*, HeLa cells) (*Guo et al., 2010*), zebrafish (*D. rerio*) (*Chew et al., 2013*), fruit fly (*D. melanogaster*) (*Dunn et al., 2013*), *Arabidopsis* (*A. thaliana*) (*Juntawong et al., 2014*), and yeast (*S. cerevisiae*) (*McManus et al., 2014*). We retrieved genome sequences and gene annotations from Ensembl v.70 and Ensembl Plants v.21 (*Flicek et al., 2012*).

Raw ribosome and RNA-seq sequencing reads underwent quality filtering using Condentri (v.2.2) (*Smeds and Künstner, 2011*) with the following settings (-hq=30 –lq=10). Adaptors described in the original publications were trimmed from filtered reads if at least five nucleotides of the adaptor sequence matched the end of each read. In zebrafish, reads from different developmental stages were pooled to improve read coverage. In all experiments, reads below 25 nucleotides were not considered. Clean ribosome short reads were filtered by mapping them to the corresponding species reference RNA (rRNA) using the Bowtie2 short-read alignment program (v. 2.1.0) (*Langmead et al., 2009*). Unaligned reads were aligned to a genomic reference genome with Bowtie2 allowing one mismatch in the first 'seed' region (the length of this region was selected according to the descriptions provided in each individual experiment). RNA-seq short reads were mapped with Tophat (v. 2.0.8) (*Kim et al., 2013*) to the corresponding reference genome. We allowed two mismatches in the alignment with the exception of zebrafish, for which we allowed three mismatches since the reads were significantly longer. Multiple mapping was allowed unless specifically stated.

### Defining a set of expressed transcripts

Expressed transcripts were assembled using Cufflinks (v 2.2.0) (*Trapnell et al., 2010*). We initially considered a transcript as expressed if it was covered by at least four reads and its abundance was higher than 1% of the most abundant isoform of the gene. We also discarded assembled transcripts in which >20% of reads were mapped to several locations in the genome. Gene annotation files from Ensembl (gtf format, v.70) were provided to Cufflinks to guide the reconstruction of already annotated transcripts. Annotated transcripts were divided into coding RNAs and long non-coding RNAs (lncRNAs), we only considered lncRNAs that were not part of genes with coding transcripts. Novel isoforms corresponding to annotated loci were not analyzed. Transcripts that did not match or overlapped annotated genes were labeled 'novel' lncRNAs. We used a length threshold of 200 nucleotides to select novel long non-coding RNAs, as in ENCODE annotations (*Djebali et al., 2012*).

Strand directionality of multiexonic transcripts was inferred using the splice site consensus sequence. We only considered monoexonic transcripts in the case of *Arabidopsis* and yeast, provided the transcripts were intergenic.

The inclusion of novel lncRNAs made it possible to perform analyses of species for which there are very few annotated lncRNAs. Annotations of UTR regions in yeast genes were missing from Ensembl because of the variability observed in transcription start sites (TSS). However, we downloaded a set of available 5′ and 3′UTRs obtained by deep transcriptomics (*Nagalakshmi et al., 2008*) and added them to the existing yeast Ensembl annotations before assembling the transcriptome.

Coding transcripts were classified into different subclasses depending on the existing annotations: (a) Annotated protein-coding transcripts (codRNA), (b) Annotated transcripts with surveillance mechanisms (nonsense mediated decay, nonstop mediated decay, and no-go decay), (c) Annotated pseudogenes. We removed protein-coding transcripts in which annotated coding sequences (CDS) are still incomplete.

Subsequently, we defined an additional subset of annotated protein-coding transcripts with well-established coding properties based on the existence of an experimentally verified protein in Swiss-Prot for the gene ('evidence at protein level', downloaded 29 October 2013, *UniProt Consortium, 2014*). These transcripts were labeled codRNAe. The rest of annotated protein-coding transcripts were abbreviated codRNAne. In zebrafish, most proteins are not yet experimentally validated; and therefore, we generated a single group.

We built a data set of human lncRNAs with described non-coding functions using data obtained from several recent reviews (*Ponting et al., 2009*; *Ulitsky and Bartel, 2013*; *Fatica and Bozzoni, 2014*). This data set included 29 different genes (*Supplementary file 2A*).

We used cufflinks to estimate the expression level of a transcript in FPKM units (Fragments Per Kilobase per total Million mapped reads). We used a threshold of >0.5 FPKM except in yeast, in which the average read coverage per transcript was much higher than in the other species and the threshold was set up at >5 FPKM. These thresholds guaranteed detection of ribosome association for the majority of expressed coding transcripts (>92%), while yielding proportions of transcripts comparable to those reported in the original papers.

## Definition of potential open reading frames (ORFs) and other transcript regions

We predicted all possible open reading frames (ORFs) in the expressed transcripts. We defined an ORF as any sequence starting with an AUG codon and finishing with a stop codon (TAA, TAG, or TGA), and at least 75 nucleotides long. This would correspond to a 24 amino acid protein, which is the size of the smallest complete human polypeptide found in genetic screen studies (*Hashimoto et al., 2001*). This ORF definition will not detect non-canonical ORFs with different start or stop codons, although these ORFs often correspond to regulatory ORFs (uORFs) in the 5′UTR region. In monoexonic transcripts (*Arabidopsis* and yeast), we considered all six possible different frames.

We also defined each transcript 5′UTR as the region between the transcription start site and the AUG codon from the left-most predicted ORF, and the 3′UTR the region from the stop codon in the right-most predicted ORF to the transcript end. UTRs with lengths below 30 nucleotides were not analyzed since ribosome reads could not be properly aligned to these regions due to their small size. Regions between two consecutive putatively translated ORFs (with ribosome profiling reads) were termed interORF. We only analyzed this region when the length of the interORF sequence in a transcript was 30 nucleotides or longer.

We defined a set of *bona fide* non-coding sequences sampled from intronic fragments. We used the introns of the genes expressed in each experiment, provided they did not overlap to any exons from other overlapping genes. We randomly selected fragments in such a way as to simulate the same size distribution as in the complete set of expressed transcripts. We performed 100 simulations of intron sampling to ensure the results were robust to the randomization process. We selected the longest ORF in each intronic fragment for the calculation of coding scores and GC content.

## Association with ribosomes and translational efficiency (TE)

We computed the number of reads overlapping each feature of interest (transcript, UTR, ORF, and interORF) using the BEDTools package (v. 2.16.2) (*Quinlan and Hall, 2010*). We only considered ribosome reads in which more than half of their length spanned the considered region. This was considered appropriate because the ribosome P-site is usually detected at the central region of the read, with only slight variations depending on the experimental setting. We set up a minimum ribosome profiling coverage of 75 nucleotides per transcript to define the transcript or transcript region (e.g., ORF) as associated with ribosomes. This is significantly longer than the length of the ribosome profiling sequencing reads (36–51 nucleotides) and is consistent with the minimum ORF length threshold.

The translational efficiency (TE) of a sequence has been previously defined as the density of ribosome profiling (RPF) reads normalized by transcript abundance (*Ingolia et al., 2009*). We calculated it by dividing the FPKM of the ribosome profiling experiment by the FPKM of the RNA-seq experiment. In transcripts, we also obtained the maximum TE by dividing the sequence in 90 nucleotide windows and selecting the window with the highest TE value.

In order to have a null model of ribosome binding against which to compare the ribosome profiling signal in codRNA and lncRNA transcripts, we extracted annotated 3′ untranslated regions (3′UTRs) from codRNAs in genes in which UTRs did not overlap with coding sequences from other transcripts, and by randomly selecting 3′UTRs with a minimum length of 30 nucleotides, we built a set of 3′UTR sequences with the same size distribution as the complete transcripts. For each species, we calculated the TE values for codRNAs, lncRNA, and 3′UTR sequences. We used the empirical distribution of TE values in the 3′UTRs to calculate the number of codRNAs and lncRNAs that showed significantly higher TE value than expected under the null model at a p < 0.05. These corresponded to TE values higher

than 0.1043 in mouse, 0.2556 in human, 0.0004 in zebrafish, 0.7164 in fruit fly, 0.1800 in *Arabidopsis*, and 0.0527 in yeast.

We defined the primary ORF in a transcript as the ORF with the largest number of RPF reads with respect to the total RPF reads covering the transcript. The rest of ORFs ≥24 amino acids associated with ribosomes were considered as well; when two or more ORFs overlapped, we selected the longest one. In ORFs, interORFs, and UTRs, we computed the TE along the whole region. For comparing the TE in different regions, we only considered transcripts in which all regions had >0.2 FPKM.

## Peptide evidence in existing proteomics databases

We downloaded all peptide sequences from the PeptideAtlas database: 338,013 human peptides (August 2013), 101,695 mouse peptides (June 2013), and 86,836 yeast peptides (March 2013). We investigated if the number of ribosome-associated protein-coding transcripts that matched the peptides in these databases varied with protein length. We omitted this analysis in zebrafish and *Arabidopsis* due to the lack of sufficiently large peptide databases. The matches were identified using BLASTP searches (v. 2.2.28+) (*Altschul et al., 1997*). We selected perfect matches only.

## Evidence of nonsense mediated decay in ORFs

We investigated how many primary ORFs may be candidates for being regulated via non-sense mediated decay (NMD) surveillance pathways, whose main function is to eliminate transcripts containing premature stop codons. We defined NMD candidates as all cases in which the stop-codon from a predicted ORF was located ≥55 nucleotides upstream of a splice junction site, provided the stop-codon was not in the terminal exon (*Scofield et al., 2007*). This mechanism is well characterized in protein-coding genes and it has been proposed as a way to degrade non-functional peptides translated in lncRNAs (*Tani et al., 2013*). Other surveillance mechanisms, such as non-stop-mediated decay or no-go decay, were not considered since all predicted ORFs finished at a stop codon, and we did not analyze RNA secondary structures.

## Defining ages of protein-coding transcripts

We utilized existing gene age classifications in human, mouse, and zebrafish (*Neme and Tautz, 2013*) to identify young gene classes: human primate-specific (~55.8 My), mouse rodent-specific (~61.7 My), human and mouse mammalian-specific (~225 My), zebrafish actinopterygii-specific (~420 My) (abbreviated fish) and metazoan (~800 My). In yeast, we used predefined genes specific to *S. cerevisiae* (1–3 My) (abbreviated *S. cerevisiae*) and the *Saccharomyces* group (~100 My) (*Ekman et al., 2007*). In *Arabidopsis*, we retrieved *Cruciferae*(*Brassicaceae*)-specific genes (20–40 My) (*Donoghue et al., 2011*). These genes are believed to have arisen primarily by de novo mechanisms, as no homologies in other species have been detected despite the fact that many closely related genomes have now been sequenced.

## Defining gene desert sequences

In humans, we obtained a set of gene desert sequences as defined in *Ovcharenko et al. (2005)*. We selected two stable and two flexible gene deserts (the definition depends on the degree of conservation in other species). They belonged to chromosome 4 (flexible located in coordinates 136,000,001–138,000,000; stable located in coordinates 180,000,001–182,000,010) that has a high number of gene deserts; and chromosome 17 (flexible located in coordinates 51,100,001–51,900,000; stable located in coordinates 69,300,001–70,000,000) that has a high gene density. We ensured that no protein-coding genes were annotated in subsequent Ensembl versions in these regions. We predicted all possible ORFs in these regions and evaluated their coding score and GC content.

## ORF coding score

The examination of nucleotide hexamer frequencies has been shown to be a powerful way to distinguish between coding and non-coding sequences (*Sun et al., 2013*; *Wang et al., 2013*). We computed one coding score (CS) per hexamer:

$$CS_{hexamer(i)} = \log\left(freq_{coding}\left(hexamer\left(i\right)\right) \middle/ freq_{non-coding}\left(hexamer\left(i\right)\right)\right).$$

The coding hexamer frequencies were obtained from the open reading frame of all transcripts in a species encoding experimentally validated proteins (except for zebrafish in which all protein-coding

transcripts were considered). The non-coding hexamer frequencies were calculated using the longest ORF in intronic regions, which were selected randomly from expressed protein-coding genes. Next, we used the following statistic to measure the coding score of an ORF:

$$CS_{ORF} = \frac{\sum_{i=1}^{i=n} CS_{hexamer(i)}}{n},$$

where $i$ is each sequence hexamer in the ORF, and $n$ the number of hexamers considered.

The hexamers were calculated in steps of three nucleotides in frame (dicodons). We did not consider the initial hexamers containing a Methionine or the last hexamers containing a STOP codon, since they are not informative. Given that all ORFs were at least 75 nucleotides long the minimum value for $n$ was 22.

We calculated other related statistics in a similar way. This included using an equiprobable hexamer distribution instead of the distribution obtained from non-coding sequences, or using codon frequencies instead of hexamer frequencies. These statistics showed somewhat lower power to distinguish between coding and non-coding sequences. As a complementary measure, we quantified the GC content in different coding and non-coding transcripts and ORFs.

## Sequence similarity searches

We employed BLASTP with an E-value cutoff of $10^{-4}$ to compare the amino acid sequences encoded by ORFs in different kinds of transcripts. We enabled SEG to mask low complexity regions in protein sequences before doing the homology searches. We also searched for homologues in the NCBI non-redundant (nr) protein database (*Pruitt et al., 2014*). BLAST sequence similarity search programs are based on gapped local alignments (*Altschul et al., 1997*).

## Analysis of single nucleotide polymorphisms

We downloaded all available single-nucleotide polymorphisms (SNPs) from dbSNP (*Sherry et al., 2001*) for human (~50 million), mouse (~64.2 million), and zebrafish (~1.3 million). We did not consider other species due to insufficient data for the analysis. We classified SNPs in ORFs as non-synonymous (PN, amino acid altering) and synonymous (PS, not amino acid altering). We computed the PN/PS ratio in each sequence data set by using the sum of PN and PS in all sequences. The estimation of PN/PS ratios of individual sequences was in general not reliable due to lack of sufficient SNP data. We obtained confidence intervals using the proportion test in R (see below).

## Statistical data analyses

The analysis of the data, including generation of plots and statistical tests, was done with R (*R Development Core Team, 2010*).

## Additional files

*Supplementary file 1* contains additional Tables and *Supplementary file 2* data subsets. The genomic coordinates of all transcripts used in this study (GTF files) and the amino acid sequences corresponding to primary ORFs in lncRNA with coding scores significant at p < 0.05 (FASTA files) are available at figshare (http://dx.doi.org/10.6084/m9.figshare.1114969).

# Acknowledgements

We acknowledge José Luis Villanueva-Cañas and Will Blevins for critical revision of the manuscript. We are grateful to Ivan Ovcharenko for advise on gene deserts. This work was funded by Ministerio de Economía y Competitividad (BFU2012-36820 and TIN2013-45732-C4-3-P) and Fundació ICREA (MMA).

# Additional information

### Funding

| Funder | Grant reference number | Author |
| --- | --- | --- |
| Ministerio de Economía y Competitividad | BFU2012-36820 | M Mar Alba |

| Funder | Grant reference number | Author |
|---|---|---|
| Ministerio de Economía y Competitividad | TIN2013-45732-C4-3-P | Xavier Messeguer |

The funders had no role in study design, data collection and interpretation, or the decision to submit the work for publication.

### Author contributions

JR-O, Conception and design, Acquisition of data, Analysis and interpretation of data, Drafting or revising the article; XM, JAS, Acquisition of data, Analysis and interpretation of data, Drafting or revising the article; MMA, Conception and design, Analysis and interpretation of data, Drafting or revising the article

## Additional files

### Supplementary files

• Supplementary file 1. Long non-coding RNAs as a source of new peptides. (**A**) Details on the number of coding transcripts associated with ribosomes. (**B**) ORF density and length in different types of transcripts. (**C**) Details on the number of non-coding transcripts associated with ribosomes. (**D**) Homology hits for ORFs. (**E**) GC content (%) in ORFs and complete sequences. (**F**) PN and PS values for different sequence subsets.

• Supplementary file 2. (**A**) Human ncRNA literature. (**B**) lncRNA homologies. (**C**) lncRNA top coding score. (**D**) Young codRNAe.

### Major datasets

The following previously published datasets were used:

| Author(s) | Year | Dataset title | Dataset ID and/or URL | Database, license, and accessibility information |
|---|---|---|---|---|
| Ingolia NT, Lareau LF, Weissman JS | 2011 | Ribosome Profiling of Mouse Embryonic Stem Cells Reveals the Complexity of Mammalian Proteomes | http://www.ncbi.nlm.nih.gov/geo/query/acc.cgi?acc=GSE30839 | Publicly available at NCBI Gene Expression Omnibus. |
| Guo H, Ingolia NT, Weissman JS, Bartel DP | 2010 | Mammalian microRNAs predominantly act to decrease target mRNA levels | http://www.ncbi.nlm.nih.gov/geo/query/acc.cgi?acc=GSE22004 | Publicly available at NCBI Gene Expression Omnibus. |
| Pauli A, Valen E, Lin MF, Garber M, Vastenhouw NL, Levin JZ, Sandelin A, Rinn JL, Regev A, Schier AF | 2011 | Comprehensive identification of long non-coding RNAs expressed during zebrafish embryogenesis | http://www.ncbi.nlm.nih.gov/geo/query/acc.cgi?acc=GSE32900 | Publicly available at NCBI Gene Expression Omnibus. |
| Chew G, Pauli A, Valen E, Schier A | 2013 | Ribosome Profiling over a Zebrafish Developmental Timecourse | http://www.ncbi.nlm.nih.gov/geo/query/acc.cgi?acc=GSE46512 | Publicly available at NCBI Gene Expression Omnibus. |
| Dunn JG, Weissman JS | 2013 | Ribosome profiling reveals pervasive and regulated stop codon readthrough in Drosophila melanogaster | http://www.ncbi.nlm.nih.gov/geo/query/acc.cgi?acc=GSE49197 | Publicly available at NCBI Gene Expression Omnibus. |
| Juntawong P, Girke T, Bailey-Serres J | 2013 | High-resolution mapping of ribosome footprints from Arabidopsis thaliana | http://www.ncbi.nlm.nih.gov/geo/query/acc.cgi?acc=GSE50597 | Publicly available at NCBI Gene Expression Omnibus. |
| McManus CJ, May GE, Spealman P, Shteyman A | 2014 | Ribosome profiling revelas post-transcriptional buffering of divergent gene expression in yeast | http://www.ncbi.nlm.nih.gov/geo/query/acc.cgi?acc=GSE52119 | Publicly available at NCBI Gene Expression Omnibus. |

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
