## [Decision Letter]

Thank you for sending your work entitled “Long non-coding RNAs as a source of new peptides” for consideration at *eLife*. Your article has been favorably evaluated by Aviv Regev (Senior editor) and 3 reviewers, one of whom is a member of our Board of Reviewing Editors.

The Reviewing editor and the other reviewers discussed their comments extensively before we reached this decision, and the Reviewing editor has assembled the following comments to help you prepare a revised submission.

This paper adds to the current active discussion on the coding potential of lncRNAs, the role of short open reading frames and the emergence of new genes. The authors use published ribosome association datasets, but use several analysis pipelines that go beyond the analysis that has previously been done with these data. However, there are two comparable published papers that do similar analysis, namely [37] and Guttman et al. 2013. While the former had suggested much translation of lncRNAs, the latter denies this, although there is some overlap of authors.

Major comments that need to be addressed by additional analyses and/clarification:

The crucial point is in how far ribosome associations are partly artifacts. The fraction of lncRNAs that the authors find to be associated with ribosomes is very large. Is this because the vast majority of transcripts actually are scanned by ribosomes, or could this observation be an artifact of the way the ribosome profiling data was analyzed? Pseudo-genes, and bona-fide human lncRNAs with known non-coding functions, were investigated, but the authors found evidence of ribosome binding in these putative negative controls, i.e. possible evidence for artifacts. This issue needs to be resolved more clearly, since the current paper should go beyond the Guttmann et al. 2013 line of arguments. It is necessary to provide a convincing demonstration that the analysis of ribosome profiling data is based on signal, not on noise. This could be done by different means, for instance by deriving null models describing what fraction of transcripts would be expected to be found associated with ribosomes if all of the ribosome profiling data was random, or by calculating otherwise a False Positive Rate or False Discovery Rate in the calling of “ribosome association” per transcript. You can also try something like the Bazzini 2014 or the Carvunis 2012 method. Another possibility is to choose a class of sequences with very low ribosomal association (maybe 3'UTRs are best) and use that as an upper bound on the false positive rate. The lower bound on the false positive rate is zero, and likely to remain there, but calculating an upper bound is something that should be added.

The claim is also made that these short and hard-to-annotate protein-coding genes look young according to protein-coding metrics and PN/PS. While plausible, it is also possible that they represent a mixture of genes of all ages combined with sequences that, while perhaps translated at some level, are not really genes in the functional sense of the word (at least not yet), and whose existence is therefore highly transient in evolutionary time. Contamination with these sequences could create the same statistical effect as having young genes. The presence of such contamination is also a critical piece of evidence in theories of how de novo protein birth occurs. This basically means that there are two interpretations of the data, both interesting, and not mutually exclusive. This needs to be better clarified. For instance the results of the BlastP search against codRNAs (supplementary file 8) and the results of the BlastX search against nr could be merged into one table or bar graph counting the number of BlastP and BlastX hits in lncRNA-noribo, lncRNA-ribo, and codRNA, separately, for each species.

It is unclear why the starved conditions (Table 1) were used in the yeast riboprofiling data. Starvation represses translation and therefore makes the data unreliable as a marker of translation. This should therefore be redone, perhaps with the rich media conditions of [36], but if this needs to be redone anyway, ideally with the much higher coverage data of Artieri & Fraser.

---

## [Author Response]

Following the editor’s recommendation we have constructed a null model for random ribosome binding based on the signal in annotated 3’UTRs. The null model can be rejected for about 90% of the lncRNAs, and a similar percentage of codRNAs, with p-value < 0.05, confirming that the signal in lncRNAs is not random. We have also reanalysed the yeast transcriptome using data from a recently published study (McManus et al., 2014). Although the main findings are similar to those reported using the original dataset, the ribosome profiling sequencing read coverage is higher and the yeast growth conditions standard, making the results more representative. We have performed homology searches with coding RNAs and lncRNAs not associated with ribosomes (in addition to lncRNAs associated with ribosomes as done previously). The results clearly show that lncRNAs display limited phylogenetic conservation when compared to coding RNAs.

We have also deposited the genomic coordinates of all transcripts used in this study and the amino acid sequences corresponding to primary ORFs in lncRNA with significant coding scores in figshare (http://dx.doi.org/10.6084/m9.figshare.1114969).

*The crucial point is in how far ribosome associations are partly artifacts. The fraction of lncRNAs that the authors find to be associated with ribosomes is very large. Is this because the vast majority of transcripts actually are scanned by ribosomes, or could this observation be an artifact of the way the ribosome profiling data was analyzed? Pseudo-genes, and bona-fide human lncRNAs with known non-coding functions, were investigated, but the authors found evidence of ribosome binding in these putative negative controls, i.e. possible evidence for artifacts. This issue needs to be resolved more clearly, since the current paper should go beyond the Guttmann et al. 2013 line of arguments. It is necessary to provide a convincing demonstration that the analysis of ribosome profiling data is based on signal, not on noise. This could be done by different means, for instance by deriving null models describing what fraction of transcripts would be expected to be found associated with ribosomes if all of the ribosome profiling data was random, or by calculating otherwise a False Positive Rate or False Discovery Rate in the calling of “ribosome association” per transcript. You can also try something like the Bazzini 2014 or the Carvunis 2012 method. Another possibility is to choose a class of sequences with very low ribosomal association (maybe 3'UTRs are best) and use that as an upper bound on the false positive rate. The lower bound on the false positive rate is zero, and likely to remain there, but calculating an upper bound is something that should be added*.

We have chosen as a null model annotated 3’UTRs from coding transcripts. The results provides strong evidence that the observed ribosome association in lncRNAs in not random and similar to codRNAs. See below the paragraph added in the manuscript text:

“In order to determine if the ribosome profiling signal in lncRNAs was different from noise, we compared ribosome density in the transcripts it to that in 3’untranslated regions (3’UTRs). More specifically, the null model consisted in a size-matched set of sequences containing randomly taken 3’UTR from annotated coding transcripts. Ribosome density was calculated as the number of ribosome profiling reads divided by RNA-seq reads, a ratio defined as Translational Efficiency (TE) (37). Both codRNAs and lncRNAS displayed much higher TE values than 3’UTRs in all species studied (Wilcoxon test p-value < 10^-5^, Figure 3). We could reject the null model for 90.12% of the lncRNAs and 87.19% of the codRNAs associated with ribosomes (p-value < 0.05) (see details by species in Table 2, Stringent set). Therefore, we concluded that the density of ribosomes in lncRNAs is much higher than expected by spurious ribosome binding.”

*The claim is also made that these short and hard-to-annotate protein-coding genes look young according to protein-coding metrics and PN/PS. While plausible, it is also possible that they represent a mixture of genes of all ages combined with sequences that, while perhaps translated at some level, are not really genes in the functional sense of the word (at least not yet), and whose existence is therefore highly transient in evolutionary time. Contamination with these sequences could create the same statistical effect as having young genes. The presence of such contamination is also a critical piece of evidence in theories of how* de novo *protein birth occurs. This basically means that there are two interpretations of the data, both interesting, and not mutually exclusive. This needs to be better clarified. For instance the results of the BlastP search against codRNAs (supplementary file 8) and the results of the BlastX search against nr could be merged into one table or bar graph counting the number of BlastP and BlastX hits in lncRNA-noribo, lncRNA-ribo, and codRNA, separately, for each species*.

Previous studies have found that lncRNAs tend to be poorly conserved across species (Guttman et al., Nature 2009; Marques and Ponting, Genome Biol. 2009; Cabili, Genes Dev. 2011). This question has been thoroughly examined in a recent paper that has dated the age of human lncRNAs using de novo assembled transcriptomes from 11 other vertebrate species (Necsulea et al., Nature 2014). The authors have reported that 81% of the human lncRNAs are not conserved beyond primates and can thus be considered “young”.

In order to further confirm this trend we have extended our initial sequence homology searches to all annotated coding transcripts in the six species studied and have compared the results obtained for putatively translated ORFs in lncRNAS to those in codRNAs. The results support the extended idea that most lncRNAs are young. For example whereas we can find only protein homologues for about 13-15% of the human and mouse lncRNAs associated with ribosomes this value is > 95% for codRNAs. Details of these searches are shown in Supplementary file 1D and Supplementary file 2B.

If we discard the lncRNAs with homologues in the other species the percentage of lncRNAs associated with ribosomes continues to be very high (mouse 80.4% with respect to 81.9%, human 40.3% with respect to 43.1%) and the coding scores of the putatively translated ORFs significantly higher than those of random ORFs (new Figure 6—figure supplement 3). Therefore our observations are essentially unaltered after filtering out the oldest lncRNAs.

The idea that some of these lncRNAs are evolutionarily transient looks plausible to us. It has been shown that the rate of loss of young genes in the Drosophila obscura group is higher than that of older genes, explaining why the number of genes remains approximately constant despite a high rate of de novo gene emergence (Palmieri and Schlotterer, 2014 *eLife*). Similarly, we can speculate that lcnRNAs probably have a high probability of being lost during evolution.

*It is unclear why the starved conditions (*Table 1*) were used in the yeast riboprofiling data. Starvation represses translation and therefore makes the data unreliable as a marker of translation. This should therefore be redone, perhaps with the rich media conditions of*
[36]*, but if this needs to be redone anyway, ideally with the much higher coverage data of Artieri & Fraser*.

The available ribosome profiling data for [4] was for *Saccharomyces* hybrids. In order to use the same species as in the original study we downloaded the *Saccharomyces cerevisiae* data from a related paper, McManus et al. (2014). Although we obtained a lower number of lncRNAs than when using the dataset from [36], the reconstructed lncRNAs were longer and thus probably more reliable. The conclusions drawn are similar to those already reported using the previous dataset.